# Research on the Prediction of Optimal Frequency for Vibration Mixing and Comparison on Initial Performance of Cold-Recycled Asphalt Emulsion Mixture

**DOI:** 10.3390/ma17164003

**Published:** 2024-08-12

**Authors:** Tian Chen

**Affiliations:** Department of Civil and Environmental Engineering and Geography Science, Ningbo University, Ningbo 315211, China; chentian@nbu.edu.cn

**Keywords:** cold-recycled asphalt emulsion mixture, vibration mixing, optimal frequency, fractal theory, forced mixing, splitting strength

## Abstract

The multicomponent cold-recycled asphalt emulsion mixture (CRAEM) has the ability of antireflection cracking between the base and the bottom surface layer, but it has secondary compaction and residual void, which is not conducive to crack resistance and fatigue performance. The application of high-frequency vibration mixing technology can reduce voids and improve crack resistance, but it is limited by the complexity of testing to determine the optimal mixing frequency. The fractal dimension of gradation is deduced by fractal theory, and the prediction model for optimal frequency is proposed. Dry, wet, freeze–thaw splitting tests, and rutting tests were employed to test the early mechanical properties of high-frequency vibration mixing specimens corresponding to different vibration accelerations, and mercury inclusion tests were utilized to compare the void distribution corresponding to the optimal mixing frequency and forced mixing, and to verify the prediction model for optimal frequency. The results indicate that the high-frequency vibration mixing technology is able to benefit the initial cracking resistance (28.1% increase), moisture stability (11.2% increase), and high-temperature stability on the macro level on the optimal frequency. Meanwhile, the void distribution structure can be optimized, reducing the proportion of harmful voids and increasing the proportion of transitional pores on the micro level. However, the freeze–thaw resistance needs to be further studied. This study reduces the number and cost of experiments to determine the optimal frequency, and provides theoretical guidance and technical support for the engineering application of the CRAEM.

## 1. Introduction

The cold-recycled asphalt emulsion mixture (CRAEM) contains complex components, including recycled asphalt particles (RAPs), original aggregate, asphalt emulsion (AE), cement, and other materials. According to the long-term detection data of pavement structures containing cold-recycled asphalt base, the residual porosity of cold-recycled asphalt pavement reaches 8–12% after compaction [1], and the wheel track path produces secondary compression deformation under traffic load [2,3,4], which makes the asphalt overlay prone to fatigue cracking (Figure 1).

The susceptibility to cracking of CRAEM is closely related to its complex composition, interfacial adhesion, and overall homogeneity. These are all directly related to the mixing uniformity, which impacts the distribution of aggregates, the formation of the overall skeleton structure, and the size distribution and compactness of voids. Additionally, there exists an intricate interplay between the demulsification process of asphalt emulsion and the hydration of cement, which is directly linked to the development of mortar strength. Ultimately, this relationship exerts a critical influence on the macroscopic mechanical properties of CRAEM.

Currently, there are two main types of mixing technology used in the production: gravitation mixing technology and FM technology [5]. Gravitation mixing technology refers to a method of mixing or blending materials using gravitational force, typically by rotating or oscillating the mixing vessel or container. The gravitation mixing technology has its advantages of simplicity and convenience, but it may not be effective for mixing materials with vastly different densities or particle sizes. The main principle of the FM technique is to employ the blades in the machine to forcefully shear, squeeze, push, and knead the mixture along a specific trajectory, resulting in uniform mixing [6]. However, during water mixing, the adhesive force generated by cement, asphalt emulsion, and aged asphalt can result in false adhesion phenomena, leading to inadequate mixing quality of cold-recycled asphalt mixtures. Then, cement cannot hydrate sufficiently, asphalt emulsion has not yet developed strength, and the particle surface would be insufficiently wrapped by the cement–asphalt emulsion mortar, thus leading to suboptimal compaction of the composite mixture. Therefore, gravitation mixing and FM technology are not suitable for CRAEM, which was proved by a previous study [7].

In the last few years, the vibration mixing (VM) technique has involved the superimposition of high-frequency vibrations onto the traditional forced mixing process. High-frequency vibration mixing technology realizes the relative displacement of material molecules through the synergistic effects of high-frequency vibration and mechanical shear to achieve the purpose of better uniformity of multicomponent materials. It is widely utilized in material science and civil engineering [8,9], and is able to improve the homogeneity of the mixtures.

High-frequency vibrations refer to elastic mechanical vibrations with frequencies close to or exceeding the low-order natural frequency of the particle system, which can induce resonance in partial particles. A characteristic of high-frequency vibrations is that higher frequencies result in faster attenuation and shorter propagation distances. Mixing with high-frequency vibrations enhances the vibrational kinetic energy of the particle system, significantly affecting particle movement patterns through energy conversion and dissipation caused by collisions. For multiple composites, with the addition of each species or a slight change of size distribution, the occurrence of segregation through VM [10] can be reduced, which is expected to reduce the complexity of RAP resources and characteristics of aged asphalt and void structure.

Meanwhile, particle size can affect the homogeneity of the mixture in VM. The intricate nature of particle trajectories observed during high-frequency vibration mixing contributes a significant role in enhancing the homogeneity of the final mixture [11]. The complex particle trajectories result in increased particle interactions and contact points, facilitating the distribution and blending of materials within the mixture. Furthermore, the high-frequency vibration mixing technique effectively disrupts the agglomeration force among fine particles, while simultaneously facilitating the centrifugal penetration of particles [12]. The bidirectional enhancement of particle movement and interaction, achieved through high-frequency vibration mixing, leads to a more comprehensive blending of asphalt binder, aggregates, and any additives or modifiers within the cold-recycled mixture. This process results in improvements in meso-homogeneity and macro-compactness, thereby improving the meso-homogeneity and macro-compactness of the mixture, and further improving the crack resistance and fatigue resistance significantly.

To date, most researchers have primarily focused on sustainable and efficient pavement construction, including aspects such as design, performance, and modeling, in the field of CRAEM [13,14,15,16,17,18,19,20,21]. They offer innovative solutions to utilize reclaimed materials effectively while reducing environmental impact. However, there has been relatively limited attention given to the mixing technology of CRAEM, not coincidentally, of which optimal frequency can only be determined from a large number of experiments, not verified by inherent properties of the mixture.

In general, the vibration frequency is inversely proportional to the particle size, that is, meaning that smaller particles have higher vibration frequencies [12]. In the process of mixing, a vibration excitation force is applied to the forced mixing shaft, causing it to rotate at a specific speed for shearing and stirring particles. When the vibration frequency of the mixing shaft matches the natural frequency of the material being mixed, oscillation occurs. Through the transmission of vibration energy to the particles in contact, the high-frequency vibration mixing technique induces a well-defined vibration amplitude within the material, thereby enhancing the overall efficiency of the vibration process [22,23]. The mixing process of multicomponent materials involves the combination of particles with varying sizes, and the vibration frequency is also influenced by this combination effect. This implies that the effective frequency of VM is not only determined by particle size but is also directly linked to the gradation of particle combinations.

The features of gradation can be described by the characteristics of fractal geometry, of which the nature is complex structures with self-similarity that cannot be described by traditional geometric shapes. The fractal is the essence of aggregate gradation [24].

The fractal dimension D is a measure that quantifies the complexity and irregularity of shapes, providing insight into the efficiency of space occupied by these shapes [25]. It helps us to describe the intricate and self-similar patterns that may be present in the system under study. The fractal dimension of asphalt mixture can be utilized for the analysis of particle distribution complexity and uniformity in pavement materials. Through the calculation of fractal dimension, one can evaluate the regularity of particle arrangement and spatial distribution uniformity. Moreover, they are all crucial aspects for understanding the internal structure of the mixture, designing a rational mix ratio, and predicting material properties [26].

This article aims to figure out the correlation between aggregate gradation and optimal frequency of the vibration mixing process. Further, the optimal mixing frequency is verified by the comparison of the initial mechanical performance of CRAEM under different high-frequency vibration mixing processes. Finally, the optimal mixing frequency is compared by microvoid characteristics and a prediction model is derived. These provide a theoretical basis for the reasonable design for mixing ratio, internal microstructure, prediction of material properties, and other key aspects, like material saving, time reduction for indoor testing, construction shortening, and technical guidance for engineering practice.

## 2. Materials

### 2.1. Raw Materials

The recycled asphalt pavement was in Xuchang, Henan province, China. The recycled asphalt particles were supplied from the upper and middle surface layer, and the maximum nominal particle size was no more than 19 mm. The main properties of RAP and virgin aggregates (VAs) are shown in Table 1.

In this study, Portland cement of P.C.42.5R grade was employed, with a percentage of 1.5%, beyond which the fatigue performance of the CRAEM is in a rapid decline [4]. The main properties of Portland cement are presented in Table 2. The selected asphalt emulsion was cationic slow-setting matrix asphalt emulsion, and its main properties are shown in Table 3.

### 2.2. Gradation

The upper base course is constructed using the CRAEM, which possesses commendable antireflective crack capabilities between the cement-stabilized macadam layer and the asphalt surface layer, thus allowing for the optimal utilization of its mechanical properties. To achieve this, a continuous medium-sized gradation was employed, with a maximum nominal particle size of 26.5 mm, as presented in Table 4. Due to the constraints imposed by the particle size of RAP, VAs (16–19 mm, <0.6 mm) were introduced to partially replace fragments of RAP. The aggregate mixing ratio adhered to the following proportions: RAP:VA:MP = 70.5:23.5:6. MP represents mineral powder, of which particle size is less than 0.075 mm; thus, the percentage in the gradation is 6%.

### 2.3. Optimal Moisture Content

Before the comparative tests, it is crucial to ensure proper compaction and achieving desired engineering properties in the final mixture. First, it is necessary to determine the optimal moisture content of mixing by the heavy-hammer tamping method, and then determine the optimal asphalt emulsion dosage using a data curve of splitting strength and porosity. Tap water was utilized for the mixing process.

At least 5 samples were allocated in the test, and each sample was 6 kg (dry weight). The amount of AE was set to 3.5%, and the desired moisture range was established as 3.5% to 5.5% in increments of 0.5% to carry out the heavy-hammer tamping test. The relationship between dry density and moisture content (Figure 2) was obtained by heavy-hammer tamping tests on base materials [27]. Based on the findings from compaction tests, it was determined that the optimum moisture content for the CRAEM was 5.0%, resulting in a maximum dry density of 2.163 kg/m^3^.

### 2.4. Optimal Asphalt Emulsion Content

Marshall specimens were prepared using the Marshall compaction equipment. The specimens were prepared at a fixed optimal moisture content and cement content. Different amounts of asphalt emulsion ranging from 3.5% to 5.5% were used, with an interval of 0.5%. The workability of the CRAEM also requires the addition of water, which is equal to the optimal moisture content multiplied by the aggregate mass minus the moisture content of AE. The applied water content changes with the amount of AE, but the optimal moisture content of the mixture remains unchanged.

After mixing, two parts (2500 g each) were used to determine the theoretical maximum relative density. The remaining were formed as 6 parallel specimens. The specimens underwent 50 cycles of compaction on each side and subsequent oven curing at 60 °C for 24 h. Then they were compacted an additional 25 times on each side while still hot, followed by cooling, demolding, and another round of oven curing at 60 °C for 24 h before performance testing.

The splitting strength and porosity were measured by the Marshall specimens obtained through compaction and curing processes. The porosity generally exhibited a decreasing trend as the dosage of asphalt emulsion increased, while the splitting strength initially demonstrated an upward trend, followed by a subsequent decline as the dosage of asphalt emulsion increased, as shown in Figure 3. Based on the results derived from the conducted experiments, it was determined that the ideal content of asphalt emulsion is 4.5% when the indirect tensile strength reaches the peak.

## 3. Experimental Methodology

### 3.1. Mix Order Design

To achieve the aggregate coating as a more complete state, asphalt emulsion requires higher mixing energy, either higher mixing amplitude or longer mixing time, so it is placed in the middle order. Previous studies of mixing sequence [7] have also shown that post-placement of cement can reduce the adverse external interface od aggregates and improve the adhesion strength. Additionally, to avoid premature breaking of asphalt emulsion, the mixing order of cement and mineral powder with asphalt emulsion should not be too early [20], so it is set as the last addition.

The mixing order follows these steps (Figure 4) [28]: All aggregates except mineral powder are mixed with water for 60 s, asphalt emulsion for 60 s, and finally, the addition of cement and mineral powder for 90 s.

### 3.2. Vibration Mixing Design

To explore the impact of VM on the performance of CRAEM under varying vibration energy (frequency), specimens were prepared using the VM and FM technologies, respectively. The content of cement and AE is identical.

The FM mixing was set as the control group. The vibration energy (vibration acceleration) was treated as a variable. Corresponding to the effective value of spindle vibration intensity (1 g, 2 g, 3 g, 4 g, and 5 g), we set the vibration frequencies to 20, 30, 40, 45, and 50 Hz, respectively. The letter g stands for gravitational acceleration, of which the default value is 9.8 m/s^2^. The physical significance of the vibration acceleration applied to the mixing shaft is that when the mixing shaft and the mixing particles contact and collide with each other, the instantaneous vibration energy transmitted is proportional to the vibration acceleration (1 g, 2 g, 3 g, 4 g, and 5 g) and also proportional to the vibration frequency (20, 30, 40, 45, and 50 Hz).

Two modes (vibratory and nonvibratory modes) are employed to differentiate mixing methods, but considering the single influential factor (vibration acceleration), the raw materials were stirred by an equipment with twin-shaft batch. The mixer used in this study was manufactured by Xuchang DETONG Vibratory Mixing Technology Co., Ltd. (Xuchang, China). The nominal volume of vibration mixer is 60 L. Two mixing shafts can rotate in counter-synchronous motion at a speed of 55 RPM. Each mixing shaft was equipped with seven interrupted and spirally mounted mixing blades. This setup is illustrated in Figure 5.

### 3.3. Technical Index

#### 3.3.1. Indirect Tensile Strength

Indirect tension tests (Figure 6) adhere to stipulations outlined in the asphalt mixture design specifications for China [29]. In the case of CRAEM, the compaction process is followed by a 48 h curing period at a temperature of 60 °C without humidity control, which is essential for achieving initial strength prior to testing. The strength measured through splitting tests correlates directly with the location of the pavement and the level of traffic load, necessitating a minimum strength requirement of 0.4 MPa, while having the potential to exceed 0.6 MPa.

#### 3.3.2. Dry–Wet Splitting Strength Ratio

The dry–wet splitting strength ratio (DWSR) pertains to the technical specifications outlined for asphalt pavement recycling as established in [30]. These specifications necessitate the conduction of two distinct testing conditions: (1) immersing half of the Marshall specimens in water for a period of 24 h, and (2) placing the remaining half in a dry indoor environment for 22 h, followed by a subsequent two-hour submersion, in order to concurrently assess the splitting strength. The prescribed minimum requirement for the DWSR stands at 75%. However, in the case of pavements expected to endure heavy traffic loads, this threshold is raised to a minimum requirement of 85%.

#### 3.3.3. Freeze–Thaw Tensile Strength Ratio

The freeze–thaw strength ratio (TSR) serves as a parameter for evaluating the moisture damage resistance of CRAEM under low-temperature conditions. During the freeze–thaw cycle, the specimen is subject to a series of specific procedures. First, the specimen is vacuum-sealed and immersed in water for a duration of 30 min. Subsequently, it is frozen at a temperature of −18 °C for 16 h, followed by submersion in water at 60 °C for 24 h. Finally, the sample is transferred to an environment at 25 °C for 2 h prior to undergoing the splitting test. The minimum required TSR is established at 70%. For pavements experiencing heavy traffic load levels, the minimum requirement for TSR is elevated to over 75%.

#### 3.3.4. Dynamic Stability

Dynamic stability serves as a key indicator for evaluating rutting deformation and assessing the high-temperature performance of CRAEM. It quantifies the number of repetitions of a standard axle load that can induce a deformation of 1 mm under representative high-temperature conditions of 60 °C, as defined in [29]. For the middle and lower layers of the pavement surface, a minimum requirement of 2000 repetitions is stipulated to ensure adequate performance.

## 4. Fractal Theory

The description of dynamic changes in fractal geometry necessitates the introduction of fractal dimension, which differs from the integer dimension found in traditional Euclidean geometry. Fractal dimension is a natural number that typically takes on fractional values (with integers indicating that Euclidean geometry is a special form of fractal geometry). It is used to quantify the irregularity and fragmentation characteristics of natural objects, denoted as *D*. A higher value for *D* indicates a greater complexity reflected by the object.

The CRAEM has a multilevel composite material system with self-similarity in the aggregate gradation. The fractal dimension *D* can be used to calculate and describe the characteristics of aggregate gradation.

According to the definition of the similar dimension of fractal theory, it can be concluded that the number of particles is proportional to the size of the particle:(1)$$N_{r}\propto r^{-D}$$
where $N_{r}$ is the number of particles whose size is not greater than in the particle group; *D* is the fractal dimension of aggregate particles.

By introducing a proportional coefficient *C*, the formula for calculating the number of particles with particle size r and maximum nominal particle size $r_{\max}$ can be obtained, respectively:(2)$$N_{r}=Cr^{-D}$$
(3)$$N_{r_{\max}}={\mathrm{Cr}}_{\max}^{-D}$$

The combination of (2) and (3) can be seen:(4)$$\frac{N_{r}}{N_{r_{\max}}}={\left(\frac{r}{r_{\max}}\right)}^{-D}$$

In engineering practice, the number of particles cannot be counted, while the particle mass is easy to obtain, and Equation (4) needs to be converted into a mass expression. Therefore, assuming that the particle size changes continuously, the particle mass with the particle size not greater than r in the particle group and the maximum nominal particle size are calculated, respectively, i.e.,
(5)$$M_{r}=\int_{0}^{r}K_{v}r^{3}\rho dN_{r}$$
(6)$$M_{r_{\max}}=\int_{0}^{r_{\max}}K_{v}r^{3}\rho dN_{r_{\max}}$$
where $M_{r}$ is the particle mass whose size is not greater than r; $M_{r_{\max}}$ is the total mass of all particles, $K_{v}$ is the volume shape factor, and ρ is the particle average density.

In conjunction with Formulas (5) and (6), the percentage of the mass of particles whose size is not greater than r (the sieve passing rate of the particle size r) can be calculated as
(7)$$P_{r}=\frac{M_{r}}{M_{r_{\max}}}=\frac{\int_{0}^{r}K_{v}r^{3}\rho dN_{r}}{\int_{0}^{r_{\max}}K_{v}r^{3}\rho dN_{r_{\max}}}=\frac{r^{3-D}}{r_{\max}^{3-D}}$$

It is shown that the mass percentage with particle size less than r is related to the particle size, the maximum nominal particle size, and the fractal dimension of the particle group.

Meanwhile, Equation (7) can be used to solve the fractal dimension of the set of particles. The proportional coefficient λ is introduced, then
(8)$$M_{r}=\lambda r^{3-D}$$

A logarithm is applied to both sides of Equation (8) to obtain
(9)$$\mathrm{lg}M_{r}=\mathrm{lg}\lambda +\left(3-D\right)\mathrm{lg}r$$

The above formula can be used as a log–log coordinate diagram according to the aggregate grading curve. The ordinate $\mathrm{lg}M_{r}$ and the abscissa lgr are linearly correlated, and the slope is K=3−D. The fractal dimension D=3−K can be obtained according to the slope.

During the high-frequency vibration mixing process for different materials of different gradations, the performance at different frequencies was compared to obtain the optimal frequency of vibration mixing. Different resources of references were searched, and the following Table 5 shows a summary.

The specific grading values are listed in Table 6 for comparison. It can be found that the nominal maximum particle size is different, and the ratio of coarse and fine aggregates is different as well. However, the universality of particle gradation can be effectively analyzed by employing the concept of gradation fractal dimension, which allows for a comprehensive characterization of various particle gradation systems. This approach enables researchers to uncover common characteristics and patterns across different gradations, providing valuable insights into the underlying principles governing particle size distribution in diverse materials.

According to the gradations of the different types of mixtures, the double logarithmic coordinate transformation is carried out to obtain the slope of the $\mathrm{lg}M_{r}$ − lgr curve, as presented in Figure 7.

For gap-graded gradation, such as SMA-13, the curve fitting accuracy is slightly worse. As for skeleton compact continuous gradation, the discrete points on the curve are close to a straight line, and the fitting accuracy of the trend line is above 98%, close to 100%.

In the case of discontinuous gradation, particularly in the context of concrete, the gradation fractal dimension can be analyzed in terms of two distinct components: the fractal dimension of coarse aggregates and the fractal dimension of fine aggregates. Due to the higher proportion of coarse aggregates and their significant role in high-frequency vibration mixing, the fractal dimension of coarse aggregates is considered representative in studying the correlation between gradation and optimal frequency. This approach allows for a comprehensive understanding of the relationship between particle size distribution and the most suitable vibration frequency in achieving optimal mechanical properties.

Subsequently, the slopes of all linear fit equations were meticulously documented and represented as K, while the fractal dimension of each mixture was computed and denoted as D=3−K. A comprehensive summary showcasing the obtained fractal dimensions and the corresponding fitting accuracy is presented in Table 7.

From the numerical point of view, there is a correlation between the two: the larger the fractal dimension, the greater the corresponding optimal frequency. The fractal dimension and the optimal vibration mixing frequency were plotted as scatter plots, and the trend lines were linearly correlated with high degree of fitting (Figure 8).

The fractal is the essence of the composition structure of the mixture particles, and the grading fractal dimension is an important index that can describe the roughness and fineness of the mixture particles. Since the optimal frequency of particle vibration is close to the resonance frequency of the particles, it is also related to the roughness and fineness of the particles, that is, there is a good correlation between the optimal frequency of particle vibration and the grading fractal dimension.

The formula for predicting the optimal frequency of high-frequency vibration mixing $f_{op}$ according to the fractal dimension of gradation *D* is as follows:(10)$$f_{op}=78.216D-151.57$$

On the contrary, when we find that the optimal frequency of particle vibration has a good correlation with the fractal dimension of gradation, the optimal frequency of particle vibration can be deduced and calculated under the premise of gradation determination of each mixture. Since the fractal dimension of the mixture is generally in the range of 2.4 to 2.7, the optimal vibration frequency can be inversely calculated between 35 and 59 Hz, which is used as a reference value, so as to reduce the number of experiments to determine the optimal frequency of particle vibration on a large scale, which not only saves time but also saves economic cost.

Moreover, fractal theory is the underlying logic, and the correlation function fitted above is generally applicable to a variety of construction and building materials, such as the concrete materials used in bridges, structures, geotechnical engineering, asphalt-based and cement-based materials utilized in subgrade and pavement maintenance, etc. It is of great reference significance to experiment and engineering application.

According to the gradation of the CRAEM, we can calculate the gradation fractal dimension (Figure 9) and the prediction of optimal mixing frequency (Formula (10)).
(11)$$D=3-0.4886=2.5114;\;f_{op}=78.216\times 2.5114-151.57=44.86\approx 45\mathrm{Hz}$$

## 5. Results and Discussion

### 5.1. Indirect Tensile Strength

As illustrated in Figure 10, the splitting strength exhibits an initially decreasing and then increasing trend with the increment in vibration frequency. Specifically, at frequencies of 20 Hz and 30 Hz, the splitting strength is lower compared to the nonvibrated mixing, and the reduction range is 3.02 to 4.97%. At 45 Hz, the strength reaches a peak of 0.721 MPa, indicating a growth of 28.1%. As the vibration frequency surpasses 45 Hz, the splitting strength declines; however, it still exceeds the control value of the splitting strength without high-frequency vibration mixing with an approximate increase rate of 9.4%.

Regarding the splitting strength, the impact of superimposed vibration frequency exhibits a dual nature. Lower vibration frequencies are unfavorable for enhancing the splitting strength, while excessively high frequencies not only hinder the strength increase but also lead to excessive energy consumption by the mixing equipment. There exists an optimal vibration mixing frequency that yields the maximum splitting strength.

### 5.2. Dry-Wet Splitting Strength Ratio

For the DWSR, the increase in vibration frequency has different effects on this index, as shown in Figure 11. When the vibration frequency is 20 Hz, the ratio experiences a decrease of 5.64%. As the vibration frequency continues to rise, the DWSR surpasses the control value. Notably, at a frequency of 40 Hz, the vibration frequency significantly improves the DWSR, which is 9.7% higher than that obtained from forced mixing. When the frequency reaches 45 Hz, the DWSR reaches 95.69%, exhibiting the largest increase of 11.2%. However, at a frequency of 50 Hz, the DWSR slightly increased by 3.0% compared with the forced mixing but remains lower than the moisture resistance corresponding to the vibration frequencies of 40 and 45 Hz.

This finding indicates that the effect of vibration frequency on dry and wet splitting strength ratio aligns with its impact on the splitting strength. While each DWSR meets the specification requirements, it is notably lower than the control value, corresponding to a frequency of 20 Hz, hence falling outside the range of effective vibration frequency selection.

Regarding the DWSR, the two vibration frequencies of 40 Hz and 45 Hz can significantly promote the splitting strength of CRAEM in a wet environment. Deviating from these frequencies, both lower and higher vibration frequencies yield slightly less favorable results in comparison. Therefore, there exists an effective range of vibration frequencies for the CRAEM, which requires comprehensive indicators to determine.

### 5.3. Freeze–Thaw Tensile Strength Ratio

The trend of TSR is opposite to that of splitting strength and DWSR, as illustrated in Figure 12. Firstly, the application of vibration frequency consistently exerts a negative effect on the TSR, resulting in reduction ranges between 7.1% and 13.4%. Even though a vibration frequency of 45 Hz corresponds to the maximum ITS and DWSR, it still yields the lowest value among the experimental group in terms of the TSR, despite meeting the technical requirements of the standard. The observed phenomenon suggests that modifying the mixing process of CRAEM through high-frequency vibration ultimately results in a reduction in freeze–thaw strength. Consequently, the mixture becomes more susceptible to moisture damage when exposed to freeze–thaw cycles. This highlights the importance of carefully considering and optimizing the mixing process to enhance the overall durability and resistance of the asphalt mixture under freeze–thaw conditions.

In future research, it is crucial to focus on investigating the mechanism behind the loss of freeze–thaw strength caused by high-frequency vibration mixing of CRAEM. It is imperative to specifically address this issue to enable better application of the mixture in the pavement structure layers of cold and frozen areas.

### 5.4. Dynamic Stability

Figure 13 illustrates that there is a notable enhancement in dynamic stability with increased vibration frequency. The inclusion of cement greatly improves the dynamic stability of the CRAEM, surpassing the standard requirements (>6000 times/mm) and exhibiting excellent resistance to rutting. When considering the driven stability index, the cold-recycled asphalt mixture subjected to vibration mixing at frequencies of 40 Hz and 45 Hz demonstrates excellent rutting resistance. In terms of rut depth, the performance corresponding to a frequency of 45 Hz is superior to that at 40 Hz.

### 5.5. Void Characteristics

Previous studies [36,37] have shown that the smaller the void size, the more favorable the crack resistance. For the MIP test, only open pores that can be invaded by mercury can be measured, while voids of the CRAEM are characterized by no gel pores and void sizes above 50 nm.

Therefore, the smallest void feature is the transition void, the proportion of the transition void is much higher than that of FM, and the proportion of the large void (harmful void) under the condition of vibration mixing at 45 Hz is 12.2% lower than that of FM, as illustrated in Figure 14. In other words, the median pore size corresponding to FM is in the region of large voids, while the median pore size corresponding to VM (45 Hz) belongs to large capillary pores.

In summary, VM at the optimal frequency (45 Hz) can make the void size distribution more uniform, which is directly proven in the effectiveness of VM in optimizing mechanical properties and void microstructure of the CRAEM, and indirectly proven in the superiority of optimal vibration mixing frequency. Combined with the characterization of mechanical properties, it can be inferred that the VM technology has the potential to enhance the void structure characteristics of the material. This, in turn, has a positive correlation with the macroscopic mechanical properties, leading to improvements in crack resistance, moisture stability, and high-temperature stability. However, it is important to note that additional measures are required to further enhance the resistance to low-temperature cracking. Therefore, further research and development efforts are necessary to strengthen the performance of the mixture under low-temperature conditions.

## 6. Conclusions

Starting from the weak link in the road performance of CRAEM, the present study focused on addressing the issue of insufficient crack resistance in CRAEM by employing high-frequency vibration mixing technology. This study aimed to validate the effectiveness of incorporating this technology in the construction and production processes of CRAEM. Through experimentation and analysis, this research aimed to establish the positive impact of high-frequency vibration mixing technology on enhancing the crack-resistance properties of CRAEM, leading to improved performance and durability. On the other hand, it investigated the influence regularity of high-frequency vibration frequencies (accelerations) on enhancing DWSR, analyzing improvements through TSR and dynamic stability. Finally, the following conclusions were drawn:High-frequency vibration mixing has a positive impact on the crack resistance, general moisture damage resistance, and rutting resistance of CRAEM, among which the initial crack resistance increased by 28.1% and moisture stability increased by 11.2%. However, it has adverse effects on freeze–thaw moisture damage resistance, reducing by 13.4%.Furthermore, the frequency utilized in high-frequency vibration mixing has an optimal range. Frequencies that are too low or too high will have a negative impact on the cracking strength and moisture damage resistance.Based on the experimental results, a vibration frequency of 45 Hz is recommended as the most effective for enhancing the splitting strength and moisture damage resistance under general conditions, which is proven by the optimization of void distribution. This frequency aligns with the design requirements for the freeze–thaw splitting strength ratio and dynamic stability.

However, further efforts are still needed to investigate the mechanism behind the negative impact of high-frequency vibration mixing on freeze–thaw moisture damage resistance and to find solutions to improve this aspect of CRAEM performance.

Overall, the findings of this research study make a significant contribution to the development and application of high-frequency vibration mixing technology in CRAEM production, providing valuable insights for optimizing the pavement performance of CRAEM.

## 7. Perspective

Further evidence for the efficiency of vibration mixing process is still needed to confirm and facilitate the widespread application of the cold-recycled asphalt mixture. Potential areas of further investigation could include the following:The effect evaluation of vibration frequency on asphalt emulsion in CRAEM under freeze–thaw conditions;The influence of VM on the distribution and orientation of asphalt aggregates in the mixture, especially in relation to their response to freeze–thaw cycles;The interaction between AE and aggregates during the VM process, particularly in terms of interfacial adhesion;The role of other factors, such as the evaluation of the CRAEM of different curing period, to better understand the curing process that takes place in the CRAEM.

By addressing these research gaps, future studies can help develop strategies and techniques to improve the freeze–thaw moisture damage resistance of CRAEM, enabling its effective use in cold and frozen regions where these conditions are prevalent.

## Figures and Tables

**Figure 1 materials-17-04003-f001:**
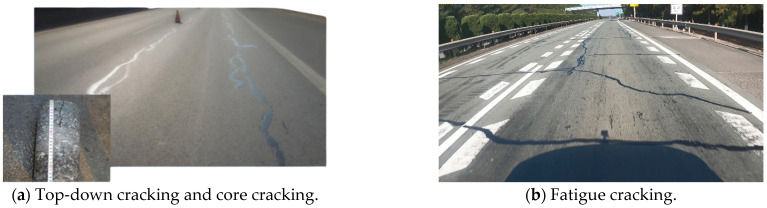
Pavement distress with CRAEM base layer.

**Figure 2 materials-17-04003-f002:**
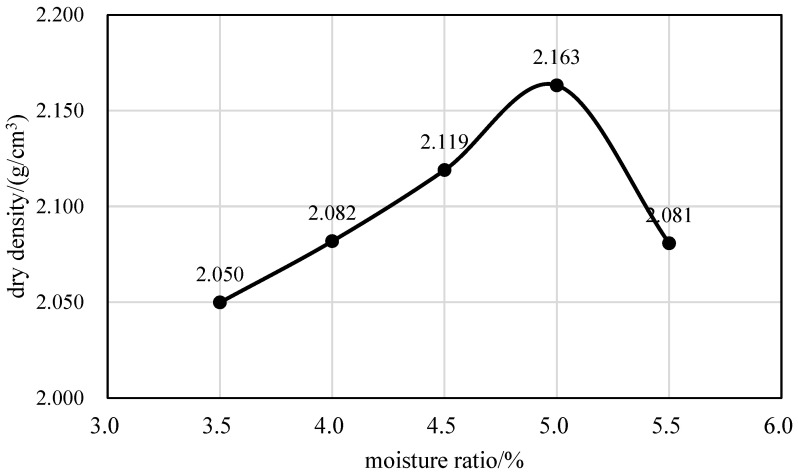
The determination of optimal moisture content.

**Figure 3 materials-17-04003-f003:**
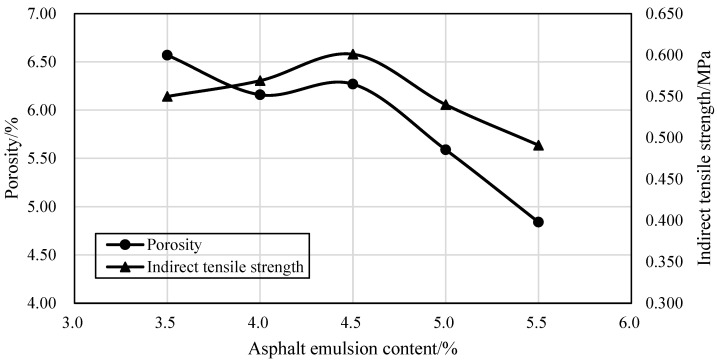
The optimum dosage of asphalt emulsion.

**Figure 4 materials-17-04003-f004:**
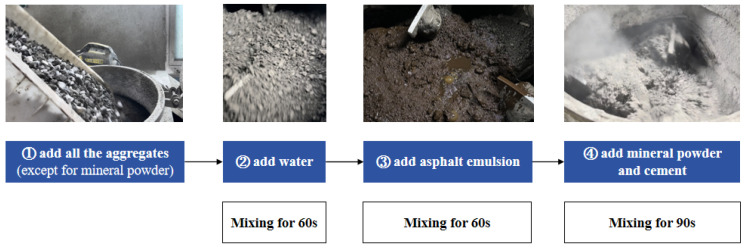
Mixing procedures of all the components.

**Figure 5 materials-17-04003-f005:**
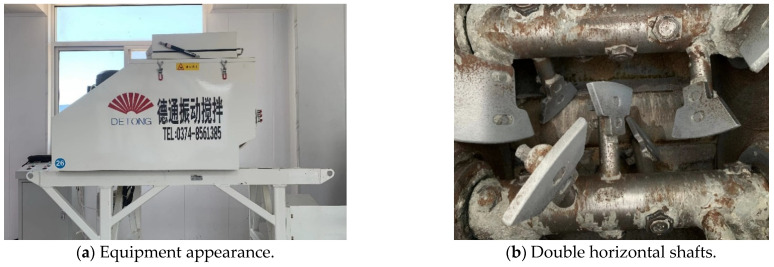
The appearance and internal structure of vibration mixing instrument.

**Figure 6 materials-17-04003-f006:**
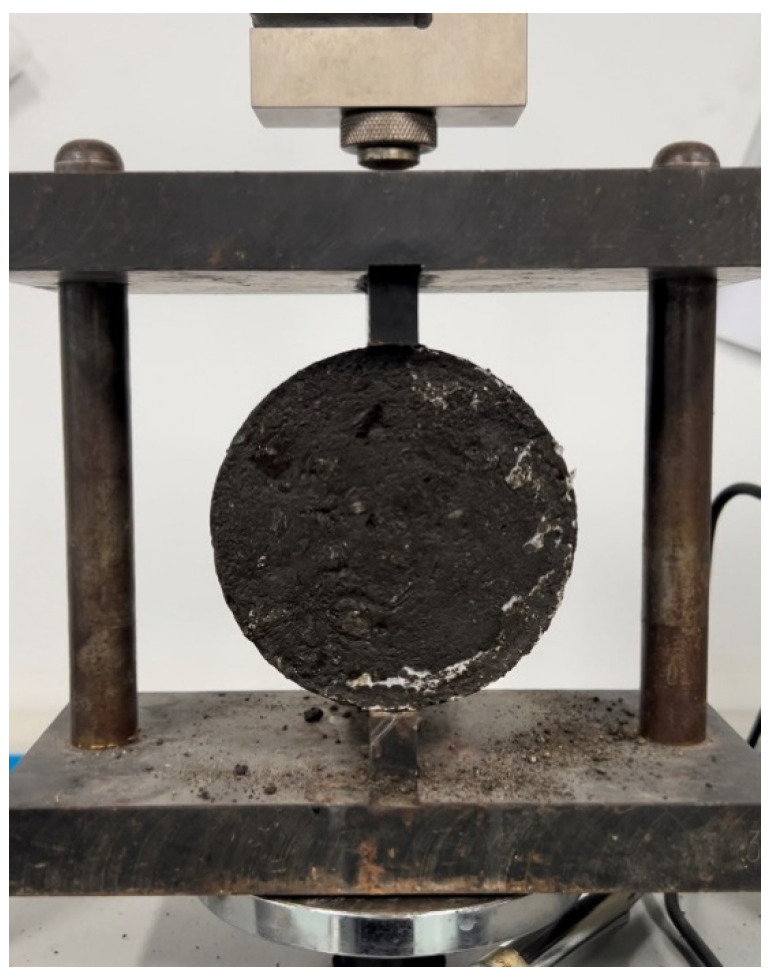
Splitting test device.

**Figure 7 materials-17-04003-f007:**
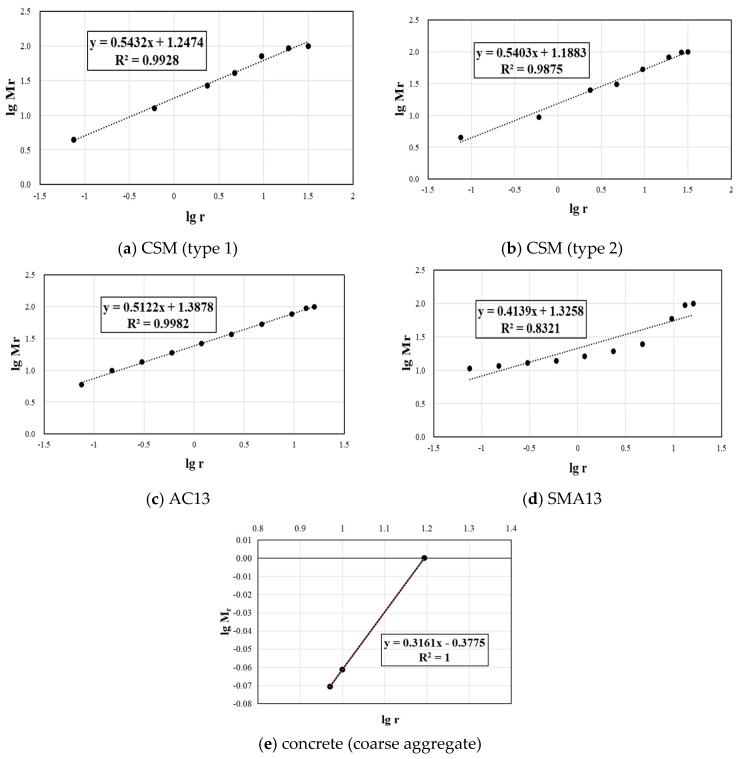
The double logarithmic coordinate diagram between gradation and particle size.

**Figure 8 materials-17-04003-f008:**
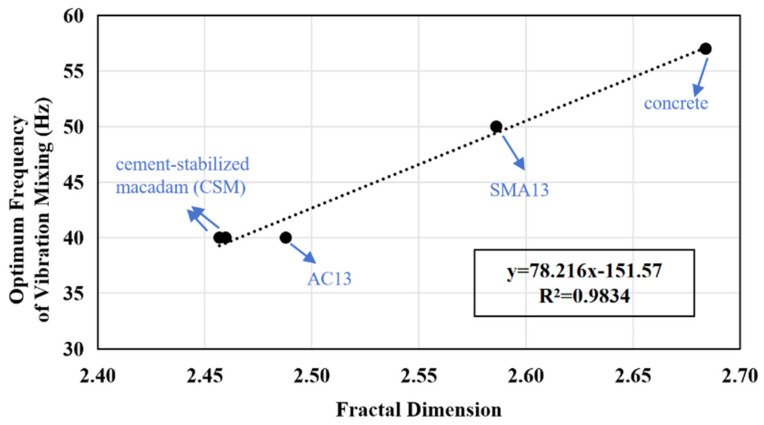
The correlation of gradation fractal dimension and optimum frequency of vibration mixing.

**Figure 9 materials-17-04003-f009:**
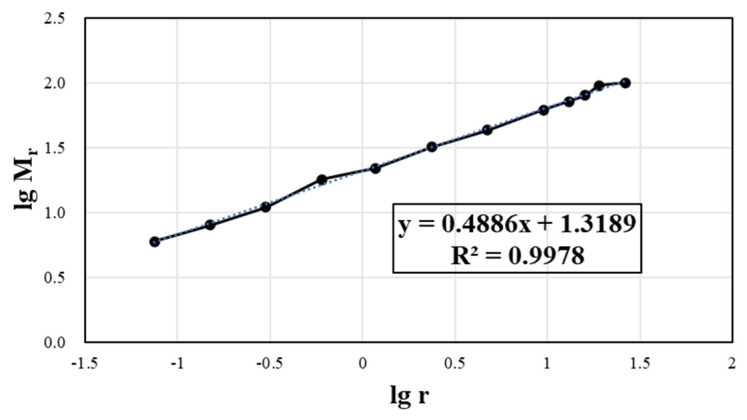
The gradation fractal dimension of CRAEM.

**Figure 10 materials-17-04003-f010:**
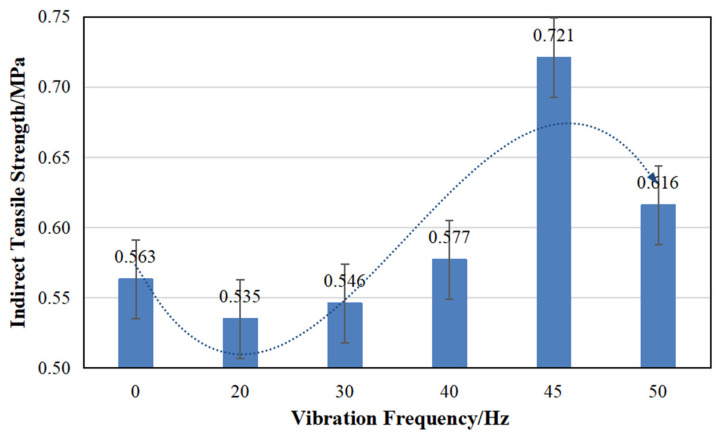
Indirect tensile strength under different vibration frequencies.

**Figure 11 materials-17-04003-f011:**
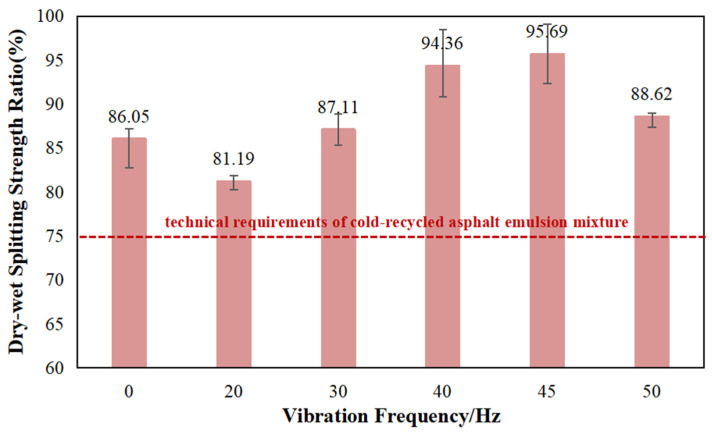
Comparative study on the moisture stability under different vibration frequencies.

**Figure 12 materials-17-04003-f012:**
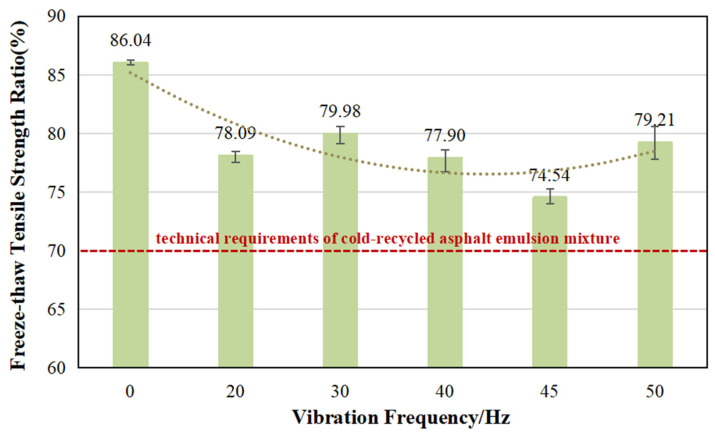
Comparison of freeze–thaw tensile strength ratio under different vibration frequencies.

**Figure 13 materials-17-04003-f013:**
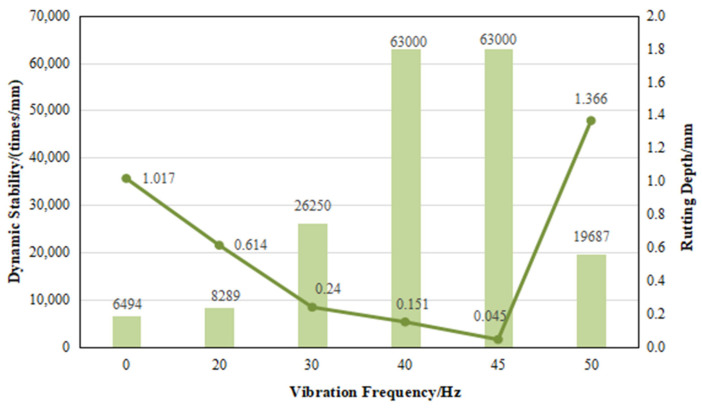
Comparative study on dynamic stability under different vibration frequencies.

**Figure 14 materials-17-04003-f014:**
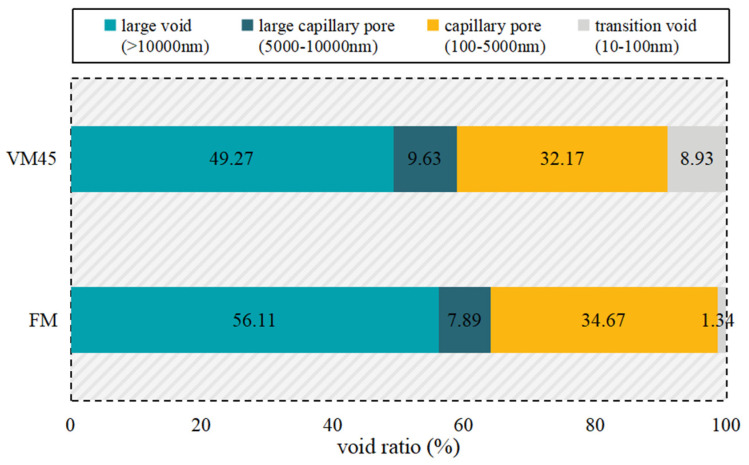
Comparison on void size distribution between FM and VM (45 Hz).

**Table 1 materials-17-04003-t001:** Main properties of RAP and VA.

Materials	Absorption (%)	Asphalt Content (%)	Density (kg/m^3^)
RAP	2.0	4.0	2400
VA	0.9	-	2700
Aged asphalt in RAP	Penetration at 25 °C (mm)	Softening point (°C)	Ductility at 15 °C (cm)
	31 × 10^−1^	57	13

**Table 2 materials-17-04003-t002:** Properties of Portland cement—42.5R grade.

Items (Unit)		Results	Requirements
Fineness (80 μm, %)		4	≤10
Initial setting time (min)		185	≥45
Final setting time (min)		247	≤600
Early strength (3d)	Compression (MPa)	28.4	≥17.0
	Bending (MPa)	5.7	≥3.5

**Table 3 materials-17-04003-t003:** Properties of asphalt emulsion (AE).

Items (Unit)	Results	Requirements
Demulsification speed	Slow-setting	Slow-setting/middle-setting
Particle charge	Cationic (+)	Cationic (+)
Residue on the 1.18 mm sieve (%)	0.02	0.1
Engler viscosity	7	2–30
Residue by distillation (%)	60	≥60
Penetration of residue (25, 0.1 mm)	74.4	50–130
Ductility of residue (15 °C, cm)	63.4	≥40
Adhesion with coarse aggregate, area ratio	≥2/3	≥2/3
Mixing state with aggregate	Uniform	Uniform
Storage stability at 1d (%)	0.2	≤1
Storage stability at 5ds (%)	1.2	≤5

**Table 4 materials-17-04003-t004:** The gradation of CRAEM.

Sieve size (mm)	26.5	19	16	13	9.5	4.75	2.36	1.18	0.6	0.3	0.15	0.075
Passing ratio (%)	100	95	80	72	62	43	32	22	18	11	8	6

**Table 5 materials-17-04003-t005:** Optimal frequency for different materials of different gradations.

Serial	Material Type	Optimal Frequency (Hz)	Gradation Type	Reference
1	Cement-stabilized macadam (CSM)	40	Continuous	[31]
2	CSM	40	Continuous	[32]
3	AC13	40	Continuous	[33]
4	SMA13	50	Gap-graded	[34]
5	concrete	57	Discontinuous	[35]

**Table 6 materials-17-04003-t006:** Gradation values for different materials.

Serial	Material Type	Passing Ratio (%)												
		31.5	26.5	19	16	13.2	9.5	4.75	2.36	1.18	0.6	0.3	0.15	0.075
1	CSM	100	-	92.3	-	-	71.3	41.2	26.9	-	12.7	-	-	4.4
2	CSM	100	98.65	81.71	-	-	53.12	30.7	24.85	-	9.43	-	-	4.52
3	AC13	-	-	-	100	95	76.5	53	37	26.5	19	13.5	10	6
4	SMA13	-	-	-	100	94.3	59.2	24.8	19.1	16.3	13.8	12.9	11.6	10.7
5	concrete	15	10	9.5	9	8	7	6	5	4.7	4	3	2.36	-
		100	86.8	85.0	79.7	67.9	53.6	39.3	21.9	15.0	44.3	5.5	0	-

**Table 7 materials-17-04003-t007:** Summary of the fractal dimension and fitting accuracy.

Serial	Material Type	Optimal Frequency (Hz)	Fractal Dimension (D)	Fitting Accuracy (%)
1	CSM	40	2.4568	99.28
2	CSM	40	2.4597	98.75
3	AC13	40	2.4878	99.82
4	SMA13	50	2.5861	83.21
5	concrete	57	2.6839	100

## Data Availability

The original contributions presented in the study are included in the article, further inquiries can be directed to the corresponding author.

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
