# Peer review of "Research on the Prediction of Optimal Frequency for Vibration Mixing and Comparison on Initial Performance of Cold-Recycled Asphalt Emulsion Mixture"

_materials, 2024, doi:10.3390/ma17164003_

Round 1

Reviewer 1 Report

Comments and Suggestions for Authors

I gladly read the paper; it covers an interesting topic that needs further exploration. From my perspective, the initial results obtained by the authors and outlined in the paper are significant, but their method must be thoroughly “strengthened” and validated with additional experimental data.

In my view, the results lack sufficient scientific evidence.

The data (and references) used to determine the formula for predicting the optimal frequency (see formula n.10) are insufficient.  Numerous studies have investigated the fractal dimension of concrete, mortar, or bitumen asphalt, thus table n.7 can be expanded to validate the work and yield a more reliable formula

To enhance the paper’s impact, I would like to suggest to consider the following aspects:

1Abstract

The abstract could be revised to be more engaging and to include, at least, some results; it seems too generic, especially from line 15 to line 20

2) Keywords

It could added “Fractal Theory” to the keywords.

3.       Section n.2

In Section 2, readers expect a more detailed description of the specimens. How many specimens were used?

4.       Acronym

Please verify the correct use of acronyms. For instance, please insert the acronym “AE” at line 138 since it’s later used in Table 3 and subsequent sections of the paper.

5.       Curve in Figure 3:

Please explain the trend observed in the curve shown in Figure 3. While there’s a description in paragraph 2.4, providing a technical rationale would be important. Why does deflection occur at an asphalt emulsion content of 4%?

6.       Points in Figure 3:

Clarify whether the points on the curve represent individual specimens or averages of multiple specimens (with the same AE percentage).

7.       Paragraph 3.1:

Please reorganized the paragraph; the authors seem to emphasize “Mixing order” over “Mix proportion design”. Some information are provided at line 202.

8.       Curing Conditions:

In the light of reproducibility principle of scientific works, greater details should be provided regarding the humidity conditions during the curing period (lines 224 and 236) of the specimens for the experimental test.

9.       Terminology:

If possible, use consistent terminology for the “material type” columns in both columns 6 and 7.

10.   Table 7:

I suggest adding the acronym “D” next to “fractional dimension.”

11.   Validity of Figure 8:

Referring to the introduction, consider the validity of the graph shown in Figure 8.

12.   Frequency Range Deduction:

Is it truly necessary to use Formula 10 to determine the optimal frequency range at line 361? Perhaps evidence from the literature review in Table 7 would suffice.

13.   State Explanation:

Clarify how the specimen used for measuring various properties at 0Hz (no vibrating mixing) was obtained and what this state represents.

Reviewer 2 Report

Comments and Suggestions for Authors

This article is interesting because it clarifies some issues related to the influence of the type of mix/compaction of cold-recycled asphalt emulsion mixture.

In this version, I found only minor details that still need to be improved or clarified:

1.      Table 2: You can add the cement type (“Properties of Portland cement - 42.5R grade”);

2.      Table 3: You can include the acronym definition “Properties of Asphalt Emulsion (AE)”;

3.      Line 152: What does the MP component (6%) mean?

4.      Subsection “2.4. Optimal Asphalt Emulsion Content”: It is not clear how the optimum water content takes into account the percentage of water included in the asphalt emulsion;

5.      Line 172: You wrote “… 50 cycles of compaction on both sides …”, but do you want to write “… 50 compaction cycles on each side …” or “… 25 compaction cycles on each side …”?

6.      Figure 4: Some images are not very clear;

7.      Line 213: Can you provide information on the average hourly output of this mixer under the conditions used?

8.      Equations (4) and (5): Where you wrote “rd”, did you mean “rd”?

9.      Table 5: The formatting of the information in column “Gradation type” leads to confusion. You can also add the names for the acronyms in the "Type of material" column;

10.   Line 323: You wrote in this line " For skeleton-extruding continuous gradation, such as SMA-13, ...". Bearing in mind that the gradation of SMA-13 is discontinuous, is this statement correct?

11.   Subsection “5.5. Void Characteristics”: How might the difference in water absorption between RAP (2%) and VA (0.9%) have affected porosity (and possibly other parameters)?

12.   In the conclusions, you can add some numbers (where appropriate);

13.   In future work (Section “7. Perspective”), it would also be interesting to evaluate some parameters in mixes of different ages (to better understand the curing process that takes place in these asphalt mixes);

14.   Some final sections have been omitted, namely Author Contributions, Conflicts of Interest, etc.

Comments on the Quality of English Language

The quality of the English Language seems adequate.

Round 2

Reviewer 1 Report

Comments and Suggestions for Authors

The revisions made this article clearer and more relevant. The information areclearly presented, and the conclusions aresupported by the data and calculations. Good job!